# *Cymbopogon citratus* Essential Oil: Its Application as an Antimicrobial Agent in Food Preservation

**Veronika Valková** [1,2] , **Hana Ďúranová** [1] , **Lucia Galovičová** [2] , **Petra Borotová** [1] , **Nenad L. Vukovic** [3] , **Milena Vukic** [3] and **Miroslava Kačániová** [2,4,*]

1   AgroBioTech Research Centre, Slovak University of Agriculture, Trieda Andreja Hlinku 2, 94976 Nitra, Slovakia; veronika.valkova@uniag.sk (V.V.); hana.duranova@uniag.sk (H.Ď.); petra.borotova@uniag.sk (P.B.)

2   Institute of Horticulture, Faculty of Horticulture and Landscape Engineering, Slovak University of Agriculture, Trieda Andreja Hlinku 2, 94976 Nitra, Slovakia; l.galovicova95@gmail.com

3   Department of Chemistry, Faculty of Science, University of Kragujevac, 34000 Kragujevac, Serbia; nvchem@yahoo.com (N.L.V.); milena.vukic@pmf.kg.ac.rs (M.V.)

4   Department of Bioenergy, Food Technology and Microbiology, Institute of Food Technology and Nutrition, University of Rzeszow, 4 Zelwerowicza Street, 35-601 Rzeszow, Poland

*   Correspondence: miroslava.kacaniova@gmail.com

**Abstract:** Antimicrobial in vitro and in situ efficacies of *Cymbopogon citratus* essential oil (lemongrass, LGEO) against 17 spoilage microorganisms (bacteria, yeasts and fungi) were evaluated. Additionally, its chemical composition, and antioxidant and antibiofilm activities were investigated. The LGEO exhibited a strong antioxidant activity (84.0 ± 0.1%), and its main constituents were citral (61.5%), geraniol (6.6%) and 1,8-cineole (6.4%). An in vitro antimicrobial evaluation revealed the lowest inhibition zone (1.00 ± 0.00 mm) in *Pseudomonas fluorescens*, and the highest inhibition zone (18.00 ± 2.46 mm) in *Candida krusei*. The values for the minimal inhibitory concentration were determined to be the lowest for *Salmonella enteritidis* and the highest for *C. albicans*. Furthermore, the concentration of ≥250 μL/L of LGEO suppressed the growth of *Penicillium aurantiogriseum*, *Penicillium expansum*, *Penicillium chrysogenum* and *Penicillium italicum*. The changes in the molecular structure of the biofilms produced by *Pseudomonas fluorescens* and *Salmonella enteritidis*, after their treatment with LGEO, confirmed its action on both biofilm-forming bacteria. Moreover, an in situ antimicrobial activity evaluation displayed the most effective inhibitory effectiveness of LGEO against *Micrococcus luteus*, *Serratia marcescens* (250 μL/L) and *Penicillium expansum* (125, 250 and 500 μL/L) growing on a carrot. Our results suggest that LGEO, as a promising natural antimicrobial agent, can be applied in the innovative packaging of bakery products and different types of vegetables, which combines commonly used packing materials with the addition of LGEO.

**Keywords:** lemongrass essential oil; chemical composition; DPPH assay; disc diffusion method; minimal inhibitory concentration; antimicrobial activity; antifungal activity; antibiofilm activity; bread; vegetable

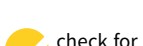



## 1. Introduction

Currently, the growing interest in healthy lifestyles and health issue concerns is strictly associated with the requirements for natural preservatives. In addition, the use of antibiotic agents causes multidrug resistance in microorganisms and, thus, an investigation of such natural constituents has attracted increased attention in the scientific community [1]. One of the possibilities for their application can be active packaging as an innovative approach, since released antimicrobial agents can minimize or eliminate the presence of food spoilage microorganisms, thereby enhancing the shelf-life of the food products [2].

It has been observed that a wide scale of valuable natural substances from different types of plants have antimicrobial properties [3]. Among them, some essential oils (EOs)

have the potential to serve as valuable preservatives of various foods, and so they can be attractive alternatives to synthetic substances [4]. Moreover, EOs are relatively easy to acquire, environmentally friendly (degrade quickly in soil and water), and they are low toxic to humans [5].

Generally, EOs are characterized as products of the different parts of a plant, containing mixtures of various chemical compounds [6], owing to their possession of a broad spectrum of biological functions, such as antioxidant, antimicrobial and anti-inflammatory qualities [7]. The extraction of EOs is carried out by two major methods, i.e., azeotropic distillation (steam distillation, hydrodiffusion and hydrodistillation) and a technique that uses solvents for extraction [8]. These secondary metabolites are mostly obtained from plants of the Poaceae [9], Pinaceae, Lamiaceae [10], Asteraceae, Rutaceae [11], Apiaceae, Lauraceae, Myrtaceae, Piperaceae and Sapindaceae angiosperm families [12], among many others. Commonly, more than 3000 EOs are known, but only about 300 of them are also commercialized [13].

Recently, there is a great interest in EOs originating from various species of *Cymbopogon* [14], which is an important genus containing more than one hundred species growing in the tropical climate zone [15]. Lemongrass (*Cymbopogon citratus*) belongs to the Poaceae family, and is among the potential EO candidates with food preserving effects [4]. Hence, the lemongrass EO (LGEO) has many biological properties that include activities, such as antibacterial [16], antifungal, antibiofilm [17] and antioxidant [18].

The chemical composition of LGEO varies depending on many factors, e.g., genetic diversity, maturity stage, light conditions, temperature or agricultural practices [19]. The lemongrass EO chemicals that are constantly detected as the constituents reflecting its biological functions are aldehydes, alcohols, hydrocarbon terpenes, esters and ketones [20]. Soliman et al. [21] found that the major compounds of LGEO were monoterpenes (96.37%), and sesquiterpenes (1.25%) and diterpenes (0.21%) were present in only small quantities. From the individual fractions, the main compound was citral (79.69% of total EO conception), which was divided into two major substances: geranial (42.86%) and neral (39.83%). Other major volatile compounds analyzed in tested EOs were myrcene (8.05%), geraniol (3.22%) and cis-verbanol (1.84%).

The main goal of the current research was to determine the in vitro antimicrobial activity of commercially obtained LGEO by using the vapor phase methods on selected food models (bread, carrot and celery) to consider its potential application as a natural agent. To perform a complex evaluation of the tested EO, the analyses of its chemical composition, antioxidant properties and antibiofilm activity were also included.

## 2. Materials and Methods

### 2.1. Essential Oil

For all analyses, a commercially available lemongrass essential oil (LGEO; *Cymbopogon citratus*) obtained by the steam distillation of fresh stalks was applied. The EO was purchased from a company (Hanus Ltd., Nitra, Slovakia) in order to complete our previous findings from such experiments [22–24]. In such a way, a comprehensive picture of the biological activities of diverse commercially available EOs purchased from the same company will be created.

### 2.2. Gas Chromatography–Mass Spectrometry

To determine the chemical profile of LGEO, an Agilent 6890N gas chromatograph coupled with a quadrupole mass spectrometer 5975B was used, as it was described by Valková et al. [22].

### 2.3. Identification of Individual Compounds

The peaks of the individual compounds were identified based on their retention indices, which were consequently compared with the library mass spectral database (Wiley and NIST databases). Using the standard method including the retention times of

n-alkanes (C6–C34) injected under the same chromatographic conditions, the retention indices were experimentally determined, and the percentage composition of compounds (relative quantity; amounts higher than 0.1%) were derived from their GC peak areas [22].

### 2.4. Antioxidant Assay

To measure the antioxidant activity (AA) of LGEO, the 2,2-diphenyl-1-picrylhydrazyl (DPPH) radical scavenging assay was used, as previously described by Galovičová et al. [23]. The AA was expressed as the percentage of DPPH inhibition, which was calculated using the following Equation (1):

$$(A0 - A1)/A0 \times 100 \tag{1}$$

where A0 was the absorbance of DPPH and A1 was the absorbance of the sample.

The power of AA was recognized as follows: weak (0–29%) < medium–strong (30–59%) < strong (60 and more %). Moreover, the value for total AA was expressed according to the calibration curve as 1 µg of the standard reference Trolox to 1 mL of the LGEO sample (TEAC). The calibration curve followed Equation (2).

$$y = -0.0006x + 0.6033$$
$$R^2 = 0.9489 \tag{2}$$

### 2.5. Tested Microorganisms

Four strains of microscopic filamentous fungi of the genus *Penicillium* (*P. expansum*, *P. italicum*, *P. aurantiogriseum*, and *P. chrysogenum*) were used to carry out the experiment. The fungi were isolated from a variety of usable materials in the food sector, and were subsequently identified using the macro- and micro-morphological characteristics based on mycological keys [25–27].

Moreover, a total of 11 bacterial strains were used in this assay: (i) four Gram-positive bacteria [*Bacillus* (*B.*) *subtilis* CCM 1999, *Enterococcus* (*E.*) *faecalis* CCM 4224, *Staphylococcus* (*S.*) *aureus* subsp. *aureus* CCM 8223 and *Micrococcus* (*M.*) *luteus* CCM 169]; (ii) three Gram-negative bacteria [*Pseudomonas* (*P.*) *aeruginosa* CCM 3955, *Serratia* (*S.*) *marcescens* CCM 8588 and *Yersinia* (*Y.*) *enterocolitica* CCM 7204]; and (iii) four yeasts [*Candida* (*C.*) *krusei* CCM 8271, *C. albicans* CCM 8261, *C. tropicalis* CCM 8223 and *C. glabrata* CCM 8270], which were obtained from the Czech collection of microorganisms (Brno, Czech Republic). In addition, two biofilm-forming bacteria including *Salmonella* (*S.*) *enteritidis* and *P. fluorescens*, were obtained from the dairy industry, and their identification was performed based on 16S rRNA gene sequence analysis and using an MALDI-TOF MS Biotyper.

### 2.6. Disc Diffusion Assay

The antimicrobial activity of LGEO was evaluated using the agar disc diffusion method [22]. To achieve this goal, an aliquot of 0.1 mL of fungal and bacterial suspension in distilled water was inoculated on Sabouraud dextrose agar (SDA; Oxoid, Basingstoke, UK) and Mueller Hinton agar (MHA; Oxoid, Basingstoke, UK), respectively. Then, the discs of filter paper (6 mm) were impregnated with 10 µL of LGEO samples (four concentrations: 62.5, 125, 250 and 500 µL/L used), and applied on the SDA and MHA surfaces. The bacteria were incubated aerobically at 37 °C for 24 h, and yeasts and fungi at 25 °C for 24 h and 5 days, respectively. The inhibition zone diameters (in mm) were measured immediately after incubation, and the power of the antimicrobial activity was expressed as follows: weak antimicrobial activity (5–10 mm) < moderate antimicrobial activity (5–10 mm) < very strong antimicrobial activity (zone > 15 mm).

### 2.7. Minimal Inhibitory Concentration

The minimal inhibitory concentration (MIC) was determined in Mueller Hinton Broth (MHB, Oxoid, Basingstoke, UK) using the broth microdilution method in 96-well polystyrene microtiter plates, according to the Galovičová et al. [23]. Prior to the experiment, the bacteria and yeasts were aerobically cultured for 24 h in MHB at 37 °C and 25 °C,

respectively. As a negative and positive control of maximal growth, the MHB and LGEO, and the MHB with inoculum were employed, respectively.

The MIC of biofilms was measured after 24 h using crystal violet, as it was reported by Kačániová et al. [24]. The absorbance of the samples was measured using a Glomax spectrophotometer (Promega Inc., Madison, WI, USA) at 570 nm. The value for the MIC was assessed based on the lowest concentration of LGEO, which completely prevented the visible growth of the tested microorganisms.

### 2.8. Antibiofilm Effect

The quantification of the in vitro biofilm production was performed according to the previously reported methodology, in the study of Kačániová et al. [24], using a MALDI-TOF MS Biotyper. *S. enteritidis* and *P. fluorescens* were used as the representatives for Gram-negative biofilm-forming bacteria. The spectra of samples were obtained using an automatic analysis, and the same sample similarities were applied to generate the standard global spectrum (MSP). In total, 19 MSPs (one planktonic spectra, 6 control spectra and 12 experimental spectra) were generated using the software MALDI Biotyper 3.0, and they were grouped into dendrograms using Euclidean distance.

### 2.9. Exposure of Food Models to the Vapor Phase of Lemongrass EO

All four fungal strains (*P. expansum*, *P. italicum*, *P. aurantiogriseum* and *P. chrysogenum*) and the two bacterial strains, *M. luteus* and *S. marcescens*, that were selected based on the results of in vitro antimicrobial activity, were used to estimate the antimicrobial activity of LGEO in situ. Here, three commonly consumed foods (wheat bread, carrot and celery) were used as the substrates for the growth of the selected microorganism species. A bakery food model was produced in the Laboratory of Cereal Technologies (Research Center AgroBioTech, SUA in Nitra), according to the methodology described in the study by Valková et al. [28]. The experiment itself was carried out according to Galovičová et al. [23]. After cooling, the slices of bread were cut to a thickness of 1.5 cm and placed into glass jars (Bormioli Rocco, Fidenza, Italy; 0.5 L). Before inoculation, bacterial and fungal strains were cultured on tryptone soya agar (TSA, Oxoid, Basingstoke, UK) for 24 h at 37 °C and on SDA at 25 °C for 5 days, respectively. Consequently, the inoculum of the tested strains was applied by stabbing the bread substrate with an injection pin three times. Then, a sterile filter paper disc (6 cm) was placed under the jar top, and 100 µL of LGEO in the concentrations of 62.5, 125, 250 and 500 µL/L (diluted in ethyl acetate) were applied to it. The control bread was not treated with LGEO. Finally, the hermetically closed jars were stored in an incubator for 7 (bacteria; 37 ± 1 °C) and 14 days (fungi; 25 ± 1 °C).

For vegetables (carrot and celery) as food models, the methodology was slightly modified. Firstly, MHA was poured into the bottom and lids of the Petri dishes (PD; 60 mm). Sliced carrot and celery (0.5 mm) were firstly placed on the agar poured at the bottom, and the inoculum was prepared as described for the bread model, but only 10 µL of LGEO (in the same concentrations) was applied on the sterile filter paper disc, and then, it was placed at the pin of PD. Subsequently, PDs were hermetically closed and cultivated at 37 °C for 7 days and at 25 °C for 14 days for bacteria and fungi, respectively.

### 2.10. Determination of Minimal Growth Inhibition

To determine the in situ bacterial and fungal growths, stereological methods were employed. Firstly, the volume densities (Vv) of microorganism colonies were estimated using ImageJ software, counting the points of the stereological grid falling into the colonies (P) and those (p) hitting the reference space (i.e., the growth substrate used: bread, carrot and celery). Consequently, each Vv of the strain colony was calculated using Equation (3). Finally, the antimicrobial activity of LGEO was expressed as the percentage of the microorganism growth inhibition (MGI), according to the Equation (4):

$$Vv \ (\%) = P/p \tag{3}$$

$$MGI = [(C - T)/C] \times 100 \qquad (4)$$

where C and T are the growth of microorganism strains (expressed as Vv) in the control and treatment group, respectively [22].

### 2.11. Statistical Analysis

A completely randomized design consisting of a minimum of three observations was conducted in the study. One-way analysis of variance (ANOVA) followed by Tukey's test at $p < 0.05$ were performed using Prism 8.0.1 (GraphPad Software, San Diego, CA, USA). To determine the values for MIC of LGEO against the growth of the selected microorganisms, logistic regression analysis was also employed.

## 3. Results

### 3.1. Chemical Composition of LGEO

The chemical constitution of LGEO was determined by GC/MS analyses. The identified volatile compounds and their percentages are presented in Table 1, in order of the highest content. As shown, 43 compounds were identified in total in LGEO, which account for 99.7% of the total volatiles. The obtained data revealed that the main component of the EO was citral (61.5%), i.e., a mixture of geranial (34.4%) and neral (27.1%), followed by geraniol (6.6%) and 1,8-cineole (6.4%).

### 3.2. Antioxidant Activity of LGEO

Using the scavenging ability of DPPH stable free radicals, we found that the values for AA of LGEO was $853.0 \pm 1.13$ TEAC (i.e., $84.0 \pm 0.1\%$), reflecting its strong antioxidant potential.

### 3.3. In Vitro Antimicrobial Properties of LGEO

The antimicrobial ability of the lemongrass EO was investigated through the disc diffusion method (Table 2). Gram-positive, Gram-negative and biofilm-forming bacteria, as well as yeasts, were used for this purpose. The results showed that LGEO had different antimicrobial activities against the growth of the microorganisms tested. It can be seen that yeasts with inhibition zones ranging from $13.33 \pm 0.58$ mm (*C. glabrata*) to $18.00 \pm 2.46$ mm (*C. krusei*) were more sensitive to LGEO than the Gram-positive and Gram-negative bacteria with inhibition zones varying between $3.67 \pm 0.58$ mm (*B. subtilis*) and $8.33 \pm 0.58$ mm (*M. luteus*), and ranging from $2.67 \pm 0.58$ mm (*S. marcescens*) to $7.68 \pm 0.58$ mm (*Y. enterocolitica*), respectively. Very weak zones of inhibition ($1.00 \pm 0.00$ mm, $2.67 \pm 0.58$ mm) were reported in the biofilm-forming bacteria, i.e., *P. fluorescens* and *S. enteritidis*, respectively.

After the preliminary screening, an MIC test was performed on all the evaluated microorganisms (Table 2). The lowest values for MIC 50 (98.21 μL/mL) and MIC 90 (112.36 μL/mL) were reported for the biofilm-forming *S. enteritidis*. By contrast, the highest MIC 50 (212.35 μL/mL) and MIC 90 (245.18 μL/mL) values were determined for *C. albicans*.

### 3.4. In Vitro Antifungal Properties of LGEO

The disc diffusion assay was also used to evaluate the sensitivity of four *Penicillium* spp. strains of the analyzed concentrations (62.5, 125, 250, and 500 μL/L) of LGEO (Table 3). It is clear from the results that the increasing concentration of LGEO enhanced its inhibitory activity against selected fungal strains; the effectiveness ($p < 0.05$) was observed in concentrations higher than 250 μL/L. More specifically, the weak values for LGEO antifungal activity were recorded against the growth of *P. aurantiogriseum* ($\geq$250 μL/L), *P. expansum* (500 μL/L), *P. chrysogenum* ($\geq$250 μL/L) and *P. italicum* (500 μL/L).

**Table 1.** Chemical composition of lemongrass essential oil.

| No | RI [a] | Compound [b] | % [c] |
|----|--------|--------------|-------|
| 1 | 909 | isobutyl isobutyrate | 0.1 |
| 2 | 926 | α-thujene | 0.1 |
| 3 | 938 | α-pinene | 1.9 |
| 4 | 948 | camphene | 0.7 |
| 5 | 977 | sabinene | 0.3 |
| 6 | 980 | β-pinene | 1.0 |
| 7 | 983 | 6-methyl-5-hepten-2-one | 1.0 |
| 8 | 992 | β-myrcene | 0.4 |
| 9 | 1016 | α-terpinene | 0.2 |
| 10 | 1117 | *p*-methyl anisole | 0.2 |
| 11 | 1023 | *p*-cimene | 0.9 |
| 12 | 1028 | α-limonene | 1.0 |
| 13 | 1033 | 1,8-cineole | 6.4 |
| 14 | 1047 | (*E*)-β-ocimene | 0.2 |
| 15 | 1060 | α-terpinene | 0.6 |
| 16 | 1087 | 4-nonanone | 0.9 |
| 17 | 1088 | α-terpinolene | 0.1 |
| 18 | 1098 | linalool | 1.7 |
| 19 | 1148 | camphor | 0.6 |
| 20 | 1152 | citronellal | 0.3 |
| 21 | 1160 | pinocarvone | 0.7 |
| 22 | 1170 | borneol | 1.2 |
| 26 | 1189 | α-terpineol | 1.5 |
| 27 | 1202 | n-decanal | 0.2 |
| 28 | 1238 | neral | 27.1 |
| 29 | 1256 | geraniol | 6.6 |
| 30 | 1266 | geranial | 34.4 |
| 31 | 1299 | geranyl formate | tr |
| 32 | 1378 | α-ylangene | 0.2 |
| 33 | 1379 | α-copaene | tr |
| 34 | 1380 | geranyl acetate | 4.3 |
| 35 | 1385 | β-bourbonene | tr |
| 36 | 1388 | β-elemene | tr |
| 37 | 1422 | (*E*)-caryophyllene | 1.8 |
| 38 | 1449 | (*E*)-isoeugenol | 0.5 |
| 39 | 1456 | α-humulene | 0.2 |
| 40 | 1483 | germacrene D | 0.2 |
| 41 | 1525 | δ-cadinene | 0.4 |
| 42 | 1542 | α-cadinene | 1.7 |
| 43 | 1566 | geranyl butanoate | 0.2 |
| | | total | 99.7 |

[a] Values of retention indices on the HP-5MS column; [b] identified compounds and [c] tr—compounds identified in amounts less than 0.1.

## 3.5. Antibiofilm Activity of LGEO

The effect of LGEO on biofilm formation by *P. fluorescens* (Figure 1, Figure 2) and *S. enteritidis* (Figure 3, Figure 4), was evaluated by MALDI-TOF MS Biotyper mass spectrometry using planktonic cells as a control group to compare biofilm molecular changes. The results revealed that the biofilm formation in the control groups (planktonic cells, as well as non-treated microorganisms) was similar (spectra not shown). Based on the mass spectra, the same peaks (Figure 1A) (indicating the same protein production) were observed between young biofilms produced by *P. fluorescens* and the control planktonic cells on day 3. At the protein level, no changes were reported in the bacterial cultures. The differences among the mass spectra of the biofilms produced on the surfaces (wood and glass) tested and the control sample were observed from day 5, with more significant dissimilarities found on the glass surface (Figure 1B–F). Alterations in the protein profile of the biofilms produced by *P. fluorescens* treated with LGEO were evident, and, thus, it can

be postulated that *Cymbopogon citratus* EO can demonstrably affect the homeostasis of the bacterial biofilm formed on wooden and glass surfaces.

**Table 2.** Antimicrobial in vitro activity of lemongrass essential oil.

| Microorganisms | Species | Zone Inhibition (mm) | Antimicrobial Effectiveness | MIC 50 (µL/mL) | MIC 90 (µL/mL) |
|---|---|---|---|---|---|
| Gram-positive bacteria | BS | 3.67 + 0.58 [af] | - | 131.24 | 163.25 |
| | EF | 4.33 ± 0.58 [ac] | - | 124.15 | 136.25 |
| | ML | 8.33 ± 0.58 [be] | * | 112.56 | 142.11 |
| | SA | 5.33 ± 1.15 [c] | * | 125.12 | 135.42 |
| Gram-negative bacteria | PA | 4.33 ± 0.58 [ac] | - | 151.25 | 174.18 |
| | YE | 7.68 ± 0.58 [e] | * | 145.18 | 156.24 |
| | SM | 2.67 ± 0.58 [f] | - | 162.18 | 181.37 |
| Yeast | CA | 13.67 ± 1.53 [g] | ** | 212.35 | 245.18 |
| | CK | 18.00 ± 2.46 [h] | *** | 196.28 | 211.36 |
| | CG | 13.33 ± 0.58 [gi] | ** | 205.26 | 221.32 |
| | CT | 16.33 ± 0.53 [j] | *** | 208.34 | 226.25 |
| Biofilm-forming bacteria | SE | 2.67 ± 0.58 [f] | - | 98.21 | 112.36 |
| | PF | 1.00 ± 0.00 [d] | - | 165.36 | 181.25 |

Note: mean ± standard deviation. Values followed by different superscripts within the same column are considerably different, statistically ($p < 0.05$). BS—*Bacillus subtilis*; EF—*Enterococcus faecalis*; ML—*Micrococcus luteus*; SA—*Staphylococcus aureus*; PA—*Pseudomonas aeruginosa*; PF—*Pseudomonas fluorescens*; YE—*Yersinia enterocolitica*; SM—*Serratia marcescens*; CA—*Candida albicans*; CK—*Candida krusei*; CG—*Candida glabrata*; CT—*Candida tropicalis*; and SE—*Salmonella enteritidis*. * Weak antimicrobial activity (zone 5–10 mm). ** Moderate inhibitory activity (zone > 10 mm). *** Very strong inhibitory activity (zone > 15 mm).

**Table 3.** Antifungal in vitro activity of lemongrass essential oil expressed by measuring the inhibition zone diameters (mm).

| Fungal Strains | LGEO (µL/L) | | | |
|---|---|---|---|---|
| | 62.5 | 125 | 250 | 500 |
| *P. aurantiogriseum* | 2.18 ± 0.26 [a] | 2.53 ± 0.41 [a] | 5.20 ± 0.50 [b*] | 7.30 ± 0.26 [c*] |
| *P. expansum* | 1.07 ± 0.21 [a] | 3.35 ± 0.37 [b] | 4.50 ± 0.57 [c] | 6.37 ± 0.15 [d*] |
| *P. chrysogenum* | 2.80 ± 0.36 [a] | 4.38 ± 0.28 [b] | 5.25 ± 0.42 [c*] | 7.10 ± 0.32 [d*] |
| *P. italicum* | 2.47 ± 0.26 [a] | 2.83 ± 0.17 [a] | 2.79 ± 0.19 [a] | 5.37 ± 0.27 [b*] |

Note: mean ± standard deviation. Values followed by different superscripts within the same row are considerably different, statistically ($p < 0.05$). * Weak antifungal activity (zone 5–10 mm).

To determine the biofilm structure similarities with respect to the standardized global spectrum (MSP) distance, a dendrogram (Figure 2) visualizing the mass spectra was designed. Here, it is evident that the planktonic stage (P), along with the control groups and young biofilms, possessed the shortest distance during the 3rd day of cultivation (PFG3, PFW3). The parallels in the protein profile of the control groups were confirmed by short MSP distances. The short MSP distances corresponding to the mass spectra were also found in the control planktonic cells and young biofilms. Another clear fact is that the distance of the experimental MSP groups has progressively increased over time. Finally, the longest MSP distances of the mass spectra analyzed on days 12 and 14 of the experiment indicate some changes in the molecular profile of *P. fluorescens*.

The kinetics of the biofilm formation by *S. enteritidis* is illustrated in Figure 3. The findings revealed that the mass spectra during days 3 and 5 (Figure 3A,B) of the *S. enteritidis* culture exhibited similar peaks for protein production in comparison with the young biofilms and control planktonic cells. Indeed, no changes in the bacterial cultures at the protein levels were observed. The differences between the mass spectra of the biofilms on the tested surfaces (wood and glass) and the control sample were evident from day 7 (Figure 3C–F). During day 9 (Figure 3D), the similarity between the planktonic spectrum and the experimental group obtained from the glass surface was observed. On the other hand, alterations were found in the protein profile of the biofilm formed by *S. enteritidis*

treated with LGEO. Thus, it can be concluded that *Cymbopogon citratus* EO can affect the homeostasis of the bacterial biofilms produced on wooden and glass surfaces.

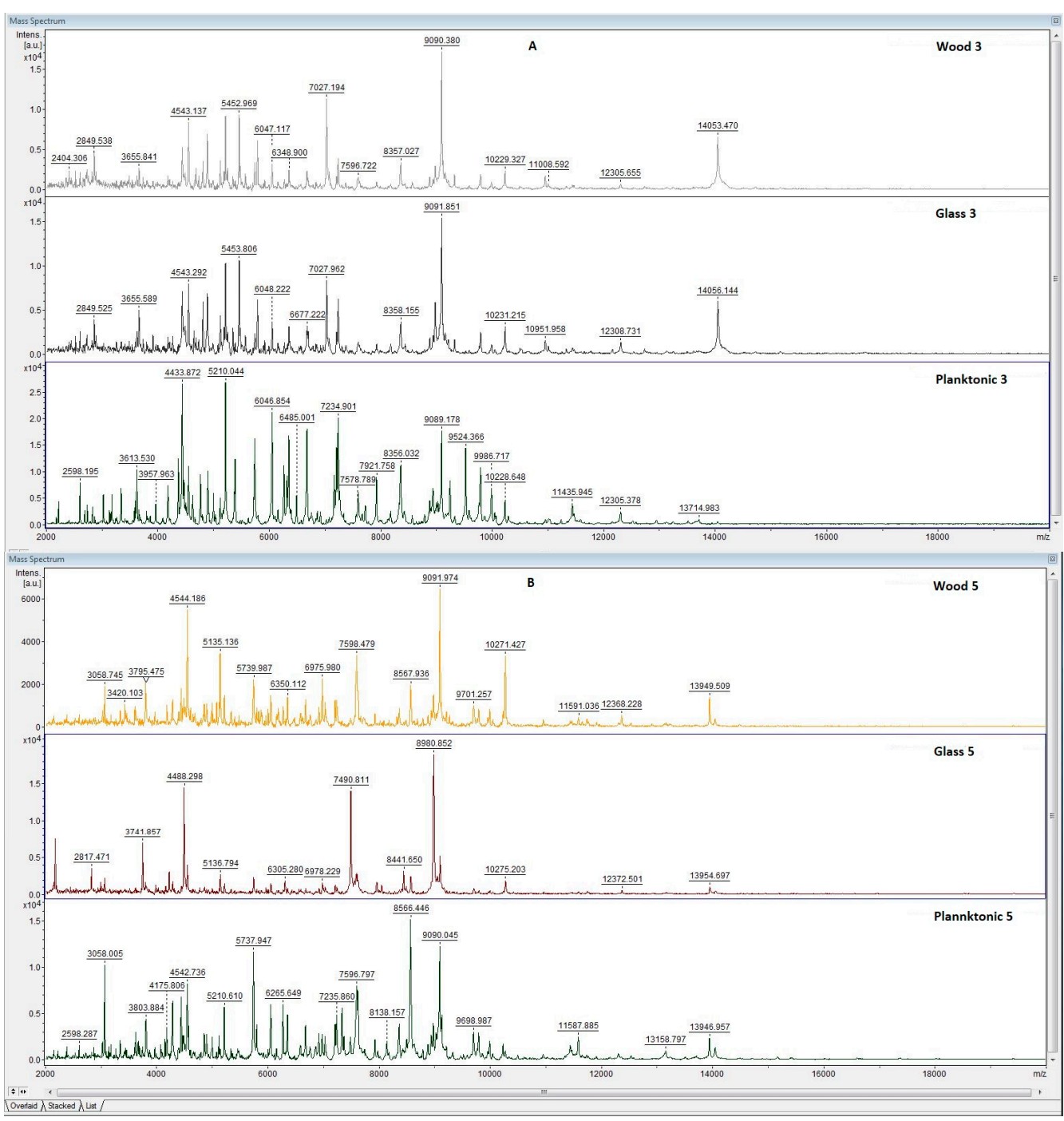

**Figure 1.** *Cont.*

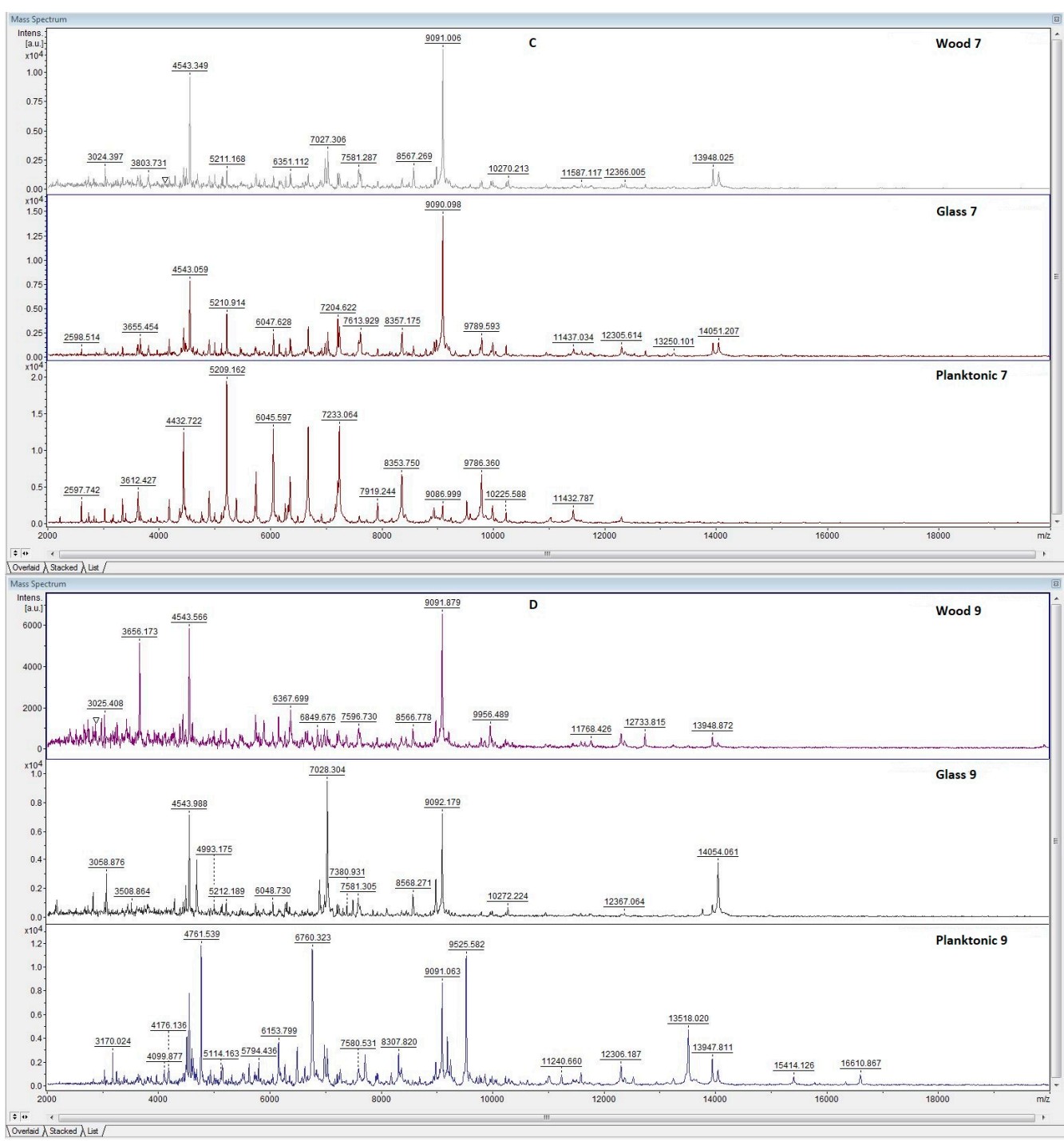

**Figure 1.** *Cont*.

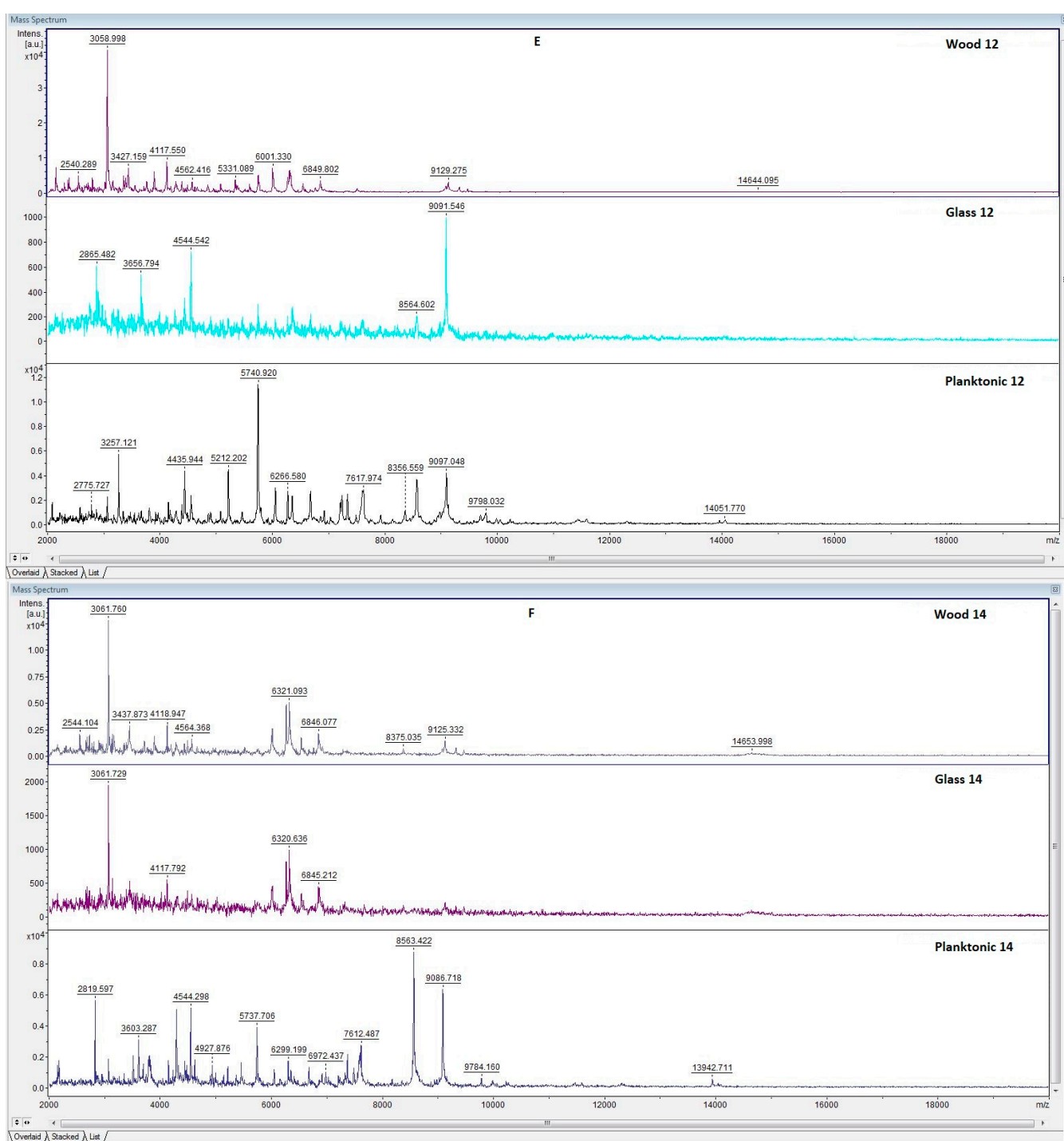

**Figure 1.** MALDI-TOF mass spectra of the biofilm produced by LGEO-treated *P. fluorescens* during development: (**A**) 3rd day, (**B**) 5th day, (**C**) 7th day, (**D**) 9th day, (**E**) 12th day and (**F**) 14th day. PF: *P. fluorescens*; C: control; G: glass; W: wood and P: planktonic.

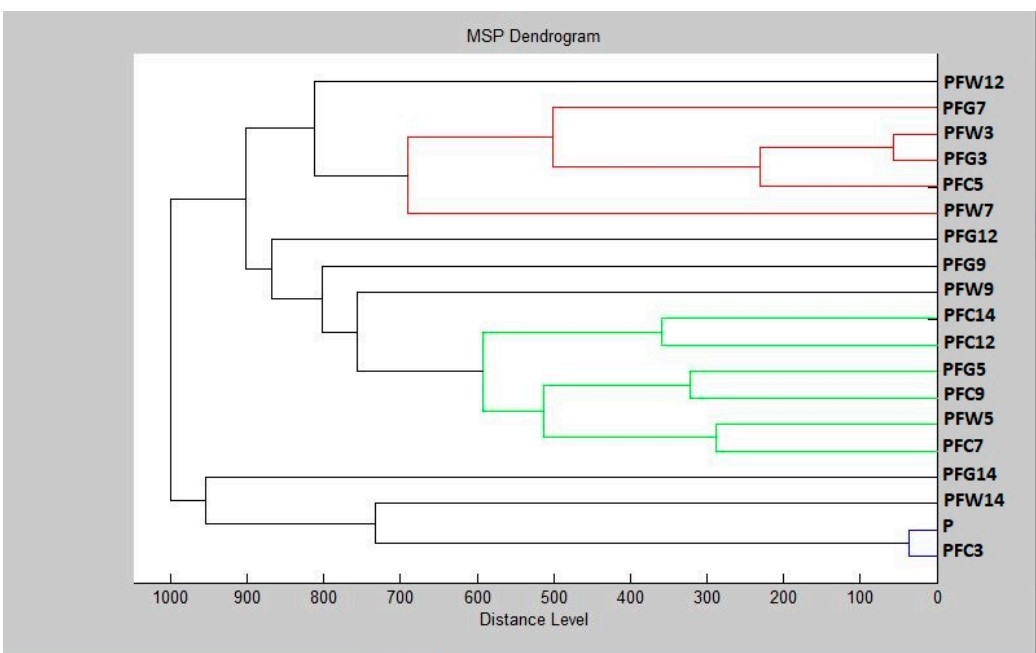

**Figure 2.** Dendrogram of *P. fluorescens* generated using the MSPs of the planktonic cells and the control. PF: *P. fluorescens*; C: control; G: glass; W: wood and P: planktonic.

Additionally, the planktonic stage (P) together with the control groups and young biofilms had the shortest distance during the 3rd and 5th days of cultivation (SEG3, SEW3, SEG5 SEW5; Figure 4). The similarity in the protein profile of the control groups was confirmed by the short MSP distances (corresponding to the mass spectra), which were also found in the young biofilms and control planktonic cells. Similar to *P. fluorescens*, the distance of the experimental MSP groups increased gradually over the time. The longest MSP distances of the mass spectra analyzed on days 9, 12 and 14 of the experiment suggest changes in the molecular profile of *S. enteritidis*.

### 3.6. In Situ Antimicrobial Properties of LGEO

The in situ antimicrobial activity evaluation revealed a strong/very strong inhibitory action of LGEO in all concentrations tested on the growth of *M. luteus* and *S. marcescens* inoculated on a bread food model. On a carrot used as a growing substrate, 250 µL/L of the EO exhibited a very strong antibacterial activity against both the selected bacteria (Table 4; Figure 5). Against *M. luteus* growing on celery, only the lowest concentration had a very strong inhibitory potential, whereas the remaining concentrations of the LGEO displayed no effectiveness. The growth of *S. marcescens* was only weakly or moderately inhibited by the application of 62.5 and 125 µL/L of LGEO, respectively.

### 3.7. In Situ Antifungal Properties of LGEO

With an increasing concentration, the lemongrass EO exhibited an enhancing inhibitory action on the growth of all four *Penicillium* spp. strains inoculated on the bread, with the strongest one in the highest concentrations (Table 5; Figure 6). The same trend was also reported in *P. aurantiogriseum* growing on the carrot as a food model. Against the growth of *P. expansum* ($\geq$125 µL/L), *P. chrysogenum* (500 µL/L) and *P. italicum* (500 µL/L) on the carrot, the EO had 100% inhibitory efficiency. On the other hand, all *Penicillium* spp. strains investigated were totally resistant to the lowest and also the lower (125 µL/L in *P. chrysogenum*; 125 and 250 µL/L in *P. italicum*) concentrations of the EO, indicating its effectiveness only at higher levels. In the case of using celery as a food model for the investigation of the in situ antifungal activity of lemongrass EO, we found that all

the *Penicillium* spp.  strains were equally highly sensitive to the actions of all the EO concentrations employed in the study.

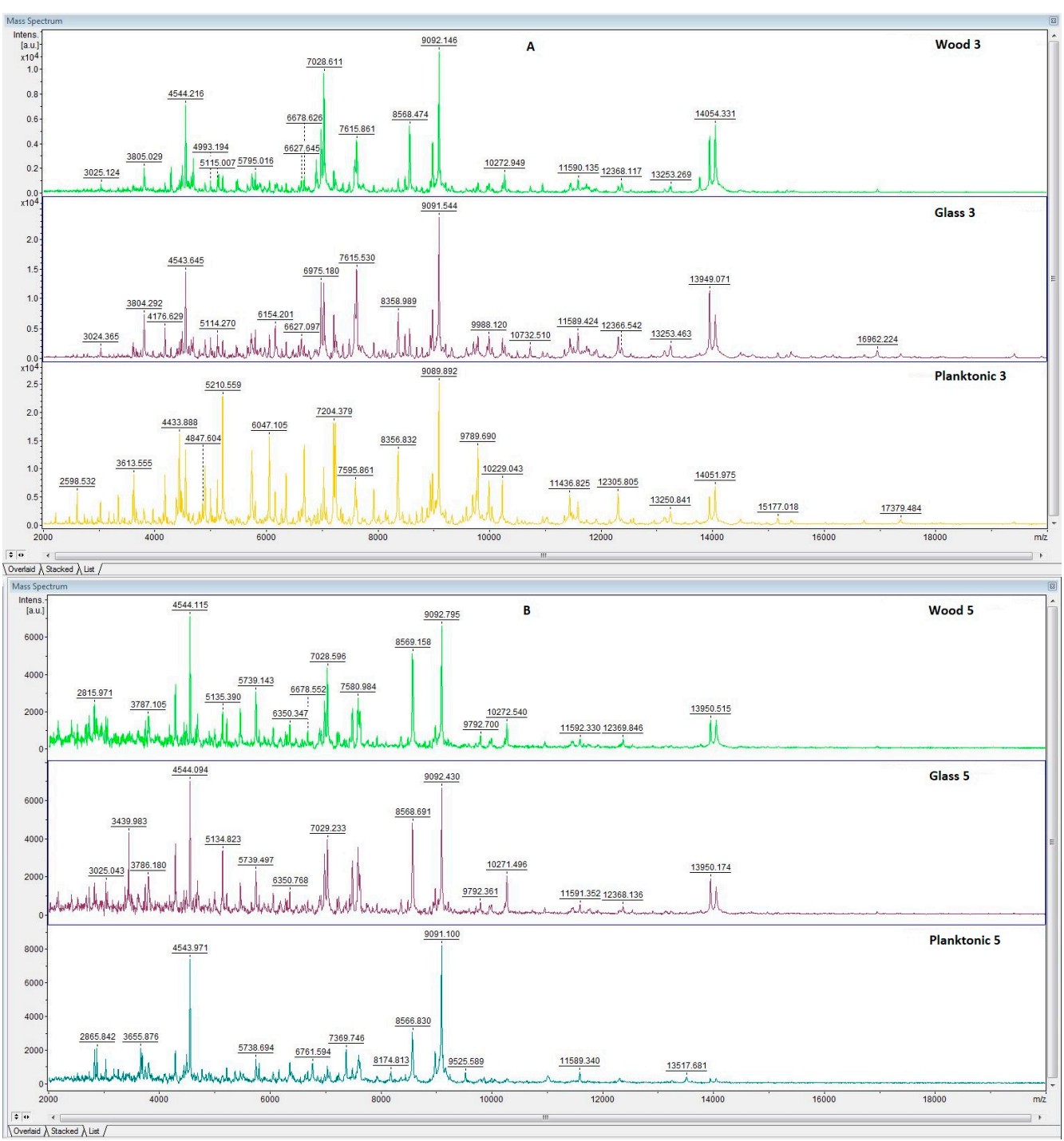

**Figure 3.** *Cont.*

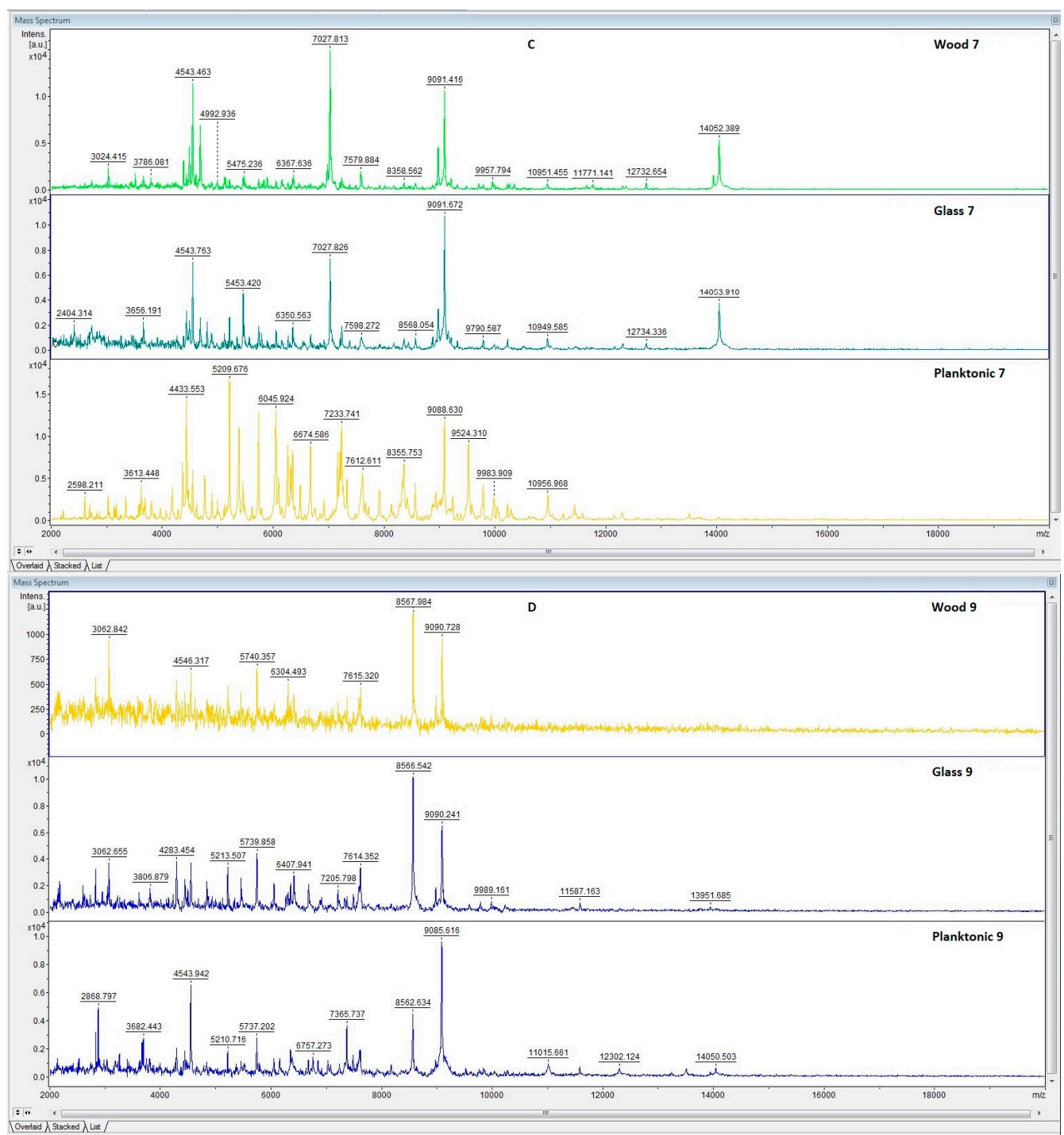

**Figure 3.** *Cont.*

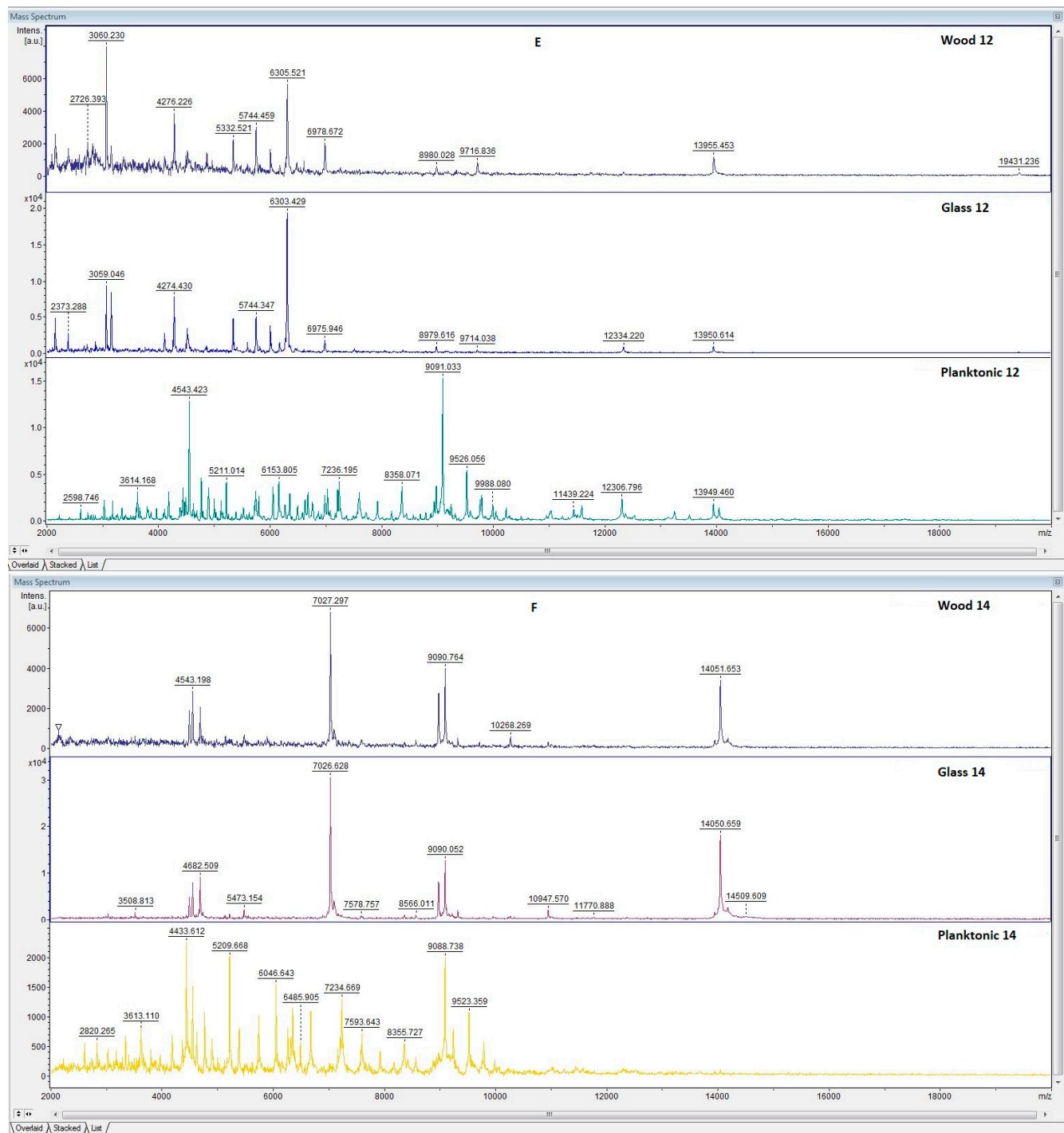

**Figure 3.** MALDI-TOF mass spectra of the biofilm produced by LGEO-treated *S. enteritidis* during development: (**A**) 3rd day, (**B**) 5th day, (**C**) 7th day, (**D**) 9th day, (**E**) 12th day and (**F**) 14th day. SE: *S. enteritidis*; C: control; G: glass; W: wood and P: planktonic.

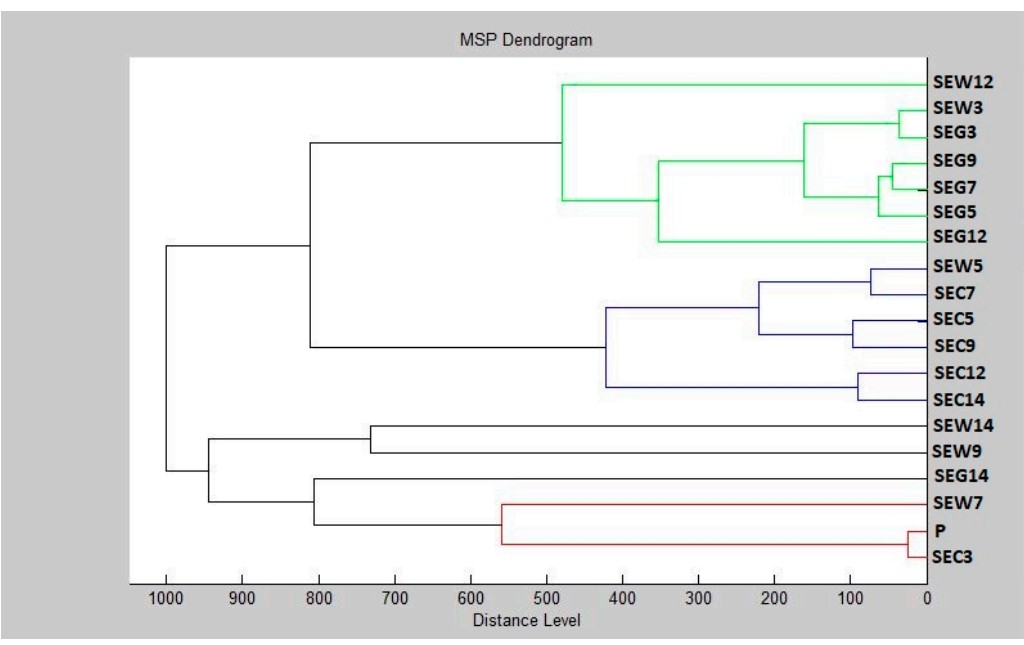

**Figure 4.** Dendrogram of *S. enteritidis* generated using the MSPs of the planktonic cells and the control. SE: S. enteritidis; C: control; G: glass; W: wood and P: planktonic.

**Table 4.** Antimicrobial in situ activity of lemongrass essential oil against two bacterial strains growing on selected food models.

| Food Model | Bacterial Strains | Bacterial Growth Inhibition (%) | | | |
|---|---|---|---|---|---|
| | | LGEO (μL/L) | | | |
| | | 62.5 | 125 | 250 | 500 |
| Bread | *M. luteus* | 88.21 ± 4.29 [a] | 90.87± 3.28 [a] | 86.77 ± 4.11 [a] | 91.49 ± 5.65 [a] |
| | *S. marcescens* | 75.33 ± 6.13 [a] | 80.50 ± 5.45 [a] | 83.24 ± 6.22 [a] | 82.35 ± 7.16 [a] |
| Carrot | *M. luteus* | 0.00 ± 0.00 [a] | 0.00 ± 0.00 [a] | 100.00 ± 0.00 [b] | 0.00 ± 0.00 [a] |
| | *S. marcescens* | 0.00 ± 0.00 [a] | 0.00 ± 0.00 [a] | 100.00 ± 0.00 [b] | 69.17 ± 6.55 [c] |
| Celery | *M. luteus* | 98.39 ± 3.32 [a] | 0.00 ± 0.00 [b] | 0.00 ± 0.00 [b] | 0.00 ± 0.00 [b] |
| | *S. marcescens* | 27.28 ± 6.14 [a] | 42.26 ± 5.91 [b] | 0.00 ± 0.00 [c] | 0.00 ± 0.00 [c] |

Note: mean ± standard deviation. Values followed by different superscripts within the same row are considerably different, statistically ($p < 0.05$).

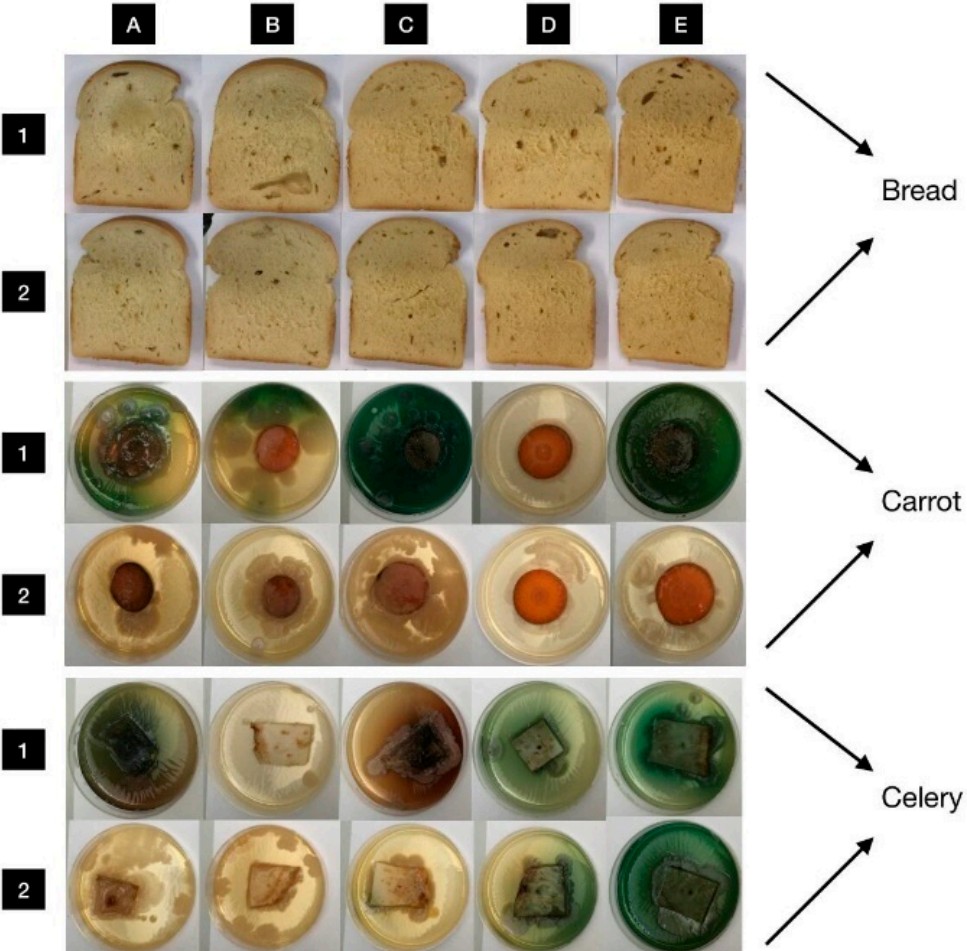

**Figure 5.** Antibacterial in situ activity of lemongrass essential oil against selected bacterial strains growing on diverse food model substrates (bread, carrot and celery samples). 1—*Micrococcus luteus*; 2—*Serratia marcescens*; (**A**)—control sample; (**B**)—62.5 μL/L; (**C**)—125 μL/L; (**D**)—250 μL/L and (**E**)—500 μL/L.

**Table 5.** Antifungal in situ activity of lemongrass essential oil against *Penicillium* spp. growing on selected food models.

| Food Model | Fungal Strains | Mycelial Growth Inhibition (%) | | | |
|---|---|---|---|---|---|
| | | LGEO (μL/L) | | | |
| | | 62.5 | 125 | 250 | 500 |
| Bread | *P. aurantiogriseum* | 44.40 ± 4.88 [a] | 84.60 ± 5.27 [b] | 87.20 ± 4.11 [b] | 95.30 ± 5.39 [b] |
| | *P. expansum* | 52.10 ± 3.88 [a] | 68.40 ± 4.17 [b] | 81.90 ± 4.89 [c] | 82.50 ± 5.99 [c] |
| | *P. chrysogenum* | 53.20 ± 5.86 [a] | 71.40 ± 4.29 [b] | 87.10 ± 4.96 [c] | 100.00 ± 0.00 [d] |
| | *P. italicum* | 65.30 ± 6.16 [a] | 87.50 ± 3.45 [b] | 85.90 ± 4.88 [b] | 89.20 ± 5.08 [b] |
| Carrot | *P. aurantiogriseum* | 0.00 ± 0.00 [a] | 11.96 ± 4.07 [b] | 52.31 ± 6.25 [c] | 100.00 ± 0.00 [d] |
| | *P. expansum* | 0.00 ± 0.00 [a] | 100.00 ± 0.00 [b] | 100.00 ± 0.00 [b] | 100.00 ± 0.00 [b] |
| | *P. chrysogenum* | 0.00 ± 0.00 [a] | 0.00 ± 0.00 [a] | 73.12 ± 4.25 [b] | 100.00 ± 0.00 [c] |
| | *P. italicum* | 0.00 ± 0.00 [a] | 0.00 ± 0.00 [a] | 0.00 ± 0.00 [a] | 100.00 ± 0.00 [b] |
| Celery | *P. aurantiogriseum* | 96.59 ± 4.33 [a] | 98.92 ± 3.12 [a] | 97.11 ± 3.41 [a] | 98.33 ± 2.64 [a] |
| | *P. expansum* | 97.25 ± 4.53 [a] | 97.87 ± 2.91 [a] | 98.26 ± 4.13 [a] | 96.45 ± 2.11 [a] |
| | *P. chrysogenum* | 97.87 ± 2.64 [a] | 96.97 ± 3.56 [a] | 95.00 ± 5.98 [a] | 98.46 ± 3.25 [a] |
| | *P. italicum* | 98.88 ± 2.18 [a] | 96.50 ± 4.26 [a] | 95.31 ± 5.23 [a] | 98.42 ± 5.27 [a] |

Note: mean ± standard deviation. Values followed by different superscripts within the same row are considerably different, statistically ($p < 0.05$).

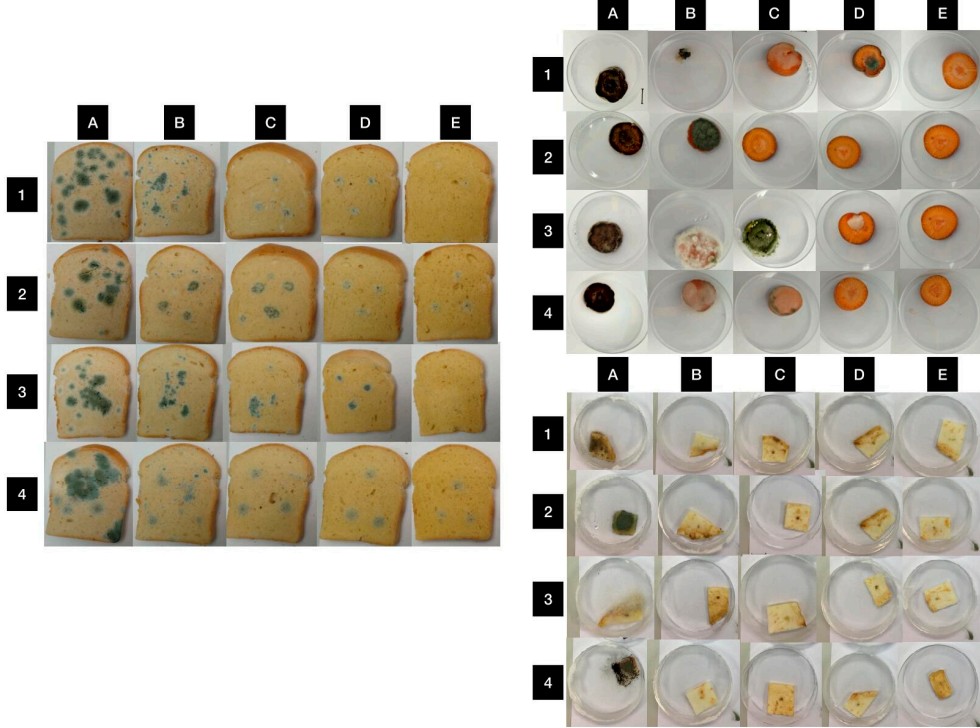

**Figure 6.** Antifungal in situ activity of lemongrass essential oil against the selected fungi strains growing on diverse food model substrates (bread, carrot and celery samples). 1—*Penicillium aurantiogriseum*; 2—*Penicillium expansum*; 3—*Penicilliumchrysogenum*; 4—*Penicillium italicum*; A—control sample; B—62.5 µL/L; C—125 µL/L; D—250 µL/L and E—500 µL/L.

## 4. Discussion

Generally, the wide range of possible applications of EOs results from the great diversity in their chemical composition [29], which can differ depending on the genetic differences, the geographical origin, the part of the plant used, the extraction method, the maturity stage or the harvest season [19]. The main constituent of *Cymbopogon citratus* EO was found to be citral, contributing to 65–80% of the total EO composition [30]. Commonly, the percentage of the substance determines the final quality of the EO [31], and produces its strong lemony aroma [32]. In our study, citral represented 61.5% of the LGEO, of which geranial and neral constituted 34.4% and 27.1% of the total volatile components, respectively. The finding corresponds to those reported by Lulekal et al. [33], Gbenou et al. [34] and Soliman et al. [21], in which citral (consisting of geranial and neral) accounted for 71.30% (40.71% and 30.59%), 46.97% (27.04% and 19.93%) and 79.69% (42.86% and 39.83%), respectively. On the contrary, the discrepancies with our results were noted by Abegaz and Yohannes [35], who detected only 13% of citral in *Cymbopogon* EO originating in Ethiopia. We hypothesize that these differences can be attributed to many factors (as described above), as well as to the time lag since the experiment of the authors was established.

Taking into account the great content of citral in our LGEO, the antibacterial and antifungal activities of the EO on food models can be expected, as it was also considered in the researches performed by Wang et al. [36] and Zhang et al. [37]. In general, the biological activities of lemongrass EO, including its antioxidant [18], antibacterial [16] and antifungal [17] potentials, have been largely examined. As it was mentioned above, the antioxidant capacity of LGEO mainly depends on its chemical composition [38], and the activity is not attributed to just one or only some of its components [39], but it is a result of the synergistic effect of all its constituents [40] with different bioactivities, functional groups and polarities. Nonetheless, recent reports indicate that citral is mainly responsible for the given value for AA since, due to its fast self-termination and cross-termination of the oxidative chain, it can cause a co-oxidation with the substrate [41]. Therefore, the high

value for the AA (853.0 ± 1.13 TEAC; 84.0 ± 0.1%) of our LGEO can also be attributed to its high proportion of citral (61.5%). In line with our findings, Mirghani et al. [42] reported a strong value for the AA (89.5%) of *Cymbopogon citratus* EO, also obtained from fresh stalks using steam distillation (similar to our oil sample). Similar results were also detected in the research conducted by Hartatie et al. [43], whose lemongrass EO that was obtained from fresh stalks by steam distillation exhibited 72.27% inhibition of radicals. On the other hand, the EO extracted by the water distillation method had lower values for AA (51.03%), reflecting the significant impact of the extraction method on the antioxidant potential of EO samples. Based on these aspects, it can be assumed that the extraction method used to obtain our LGEO was sufficient to maintain the strong antioxidant potential of the plant. Considering the strong AA of our tested EO, its antimicrobial and antifungal properties were determined in the next steps.

Since ancient times, various types of EOs have been known for their antimicrobial properties [44], whereas these secondary metabolites are capable of inhibiting or to slowing down the growth of diverse food spoilage microorganisms, such as bacteria, yeasts and microscopic filamentous fungi [45]. Currently, the antibacterial activity of lemongrass EO has been reported by many authors [46,47], and it is again mainly associated with the presence of citral. In effect, its individual constituents (i.e., neral and geranial) have an ability to exert the antibacterial action on Gram-negative and Gram-positive microorganisms [48]. However, other substances, including geraniol [49] and 1,8-cineole [50], which were also determined in our EO, were shown to exert their growth inhibition activities against various bacteria species, too. We generally propose that the chemical composition of our EO predicts its potential to inhibit the growth of microorganisms. Indeed, the underlying mechanism of LGEO antibacterial action is based on the interaction between its main components and the bacterial cell membrane. For instance, the antimicrobial activity of citral against *Cronobacter sakazakii* was linked to the alterations in the ATP concentration, hyperpolarization of the cell membrane and the decreased cytoplasmic pH [51]. In our study, LGEO displayed the highest inhibitory efficacy against the tested *Candida* strains. The antimicrobial activity of LGEO against five *Candida* species (*C. albicans*, *C. parapsilosis*, *C. tropicalis*, *C. glabrata* and *C. krusei*) was also evaluated by Silva et al. [52]. The authors observed that all of the strains were sensitive to LGEO, applied in a volume of 2.0 μL on a filter paper disc with the following inhibition zones: 14.7 mm (*C. albicans*), 19.3 mm (*C. glabrata*), 12.3 mm (*C. krusei*), 8.9 mm (*C. parapsilosis*) and 12.8 mm (*C. tropicalis*), which is consistent with our results. The finding was also confirmed by Soares et al. [53], who revealed the effectiveness of LGEO against the growth of *C. albicans*, *C. parapsilosis* and *C. tropicalis*, suggesting a new perspective of the EO application as a potential agent against *Candida* spoilage. Additionally, many studies proved the antimicrobial activity of LGEO against various bacterial strains. In effect, Naik et al. [54] analyzed the antibacterial properties of LGEO against three Gram-positive (*S. aureus*, *B. cereus* and *B. subtilis*) and three Gram-negative (*Escherichia coli*, *Klebsiella pneumoniae* and *P. aeruginosa*) bacteria, revealing that LGEO was more effective against Gram-positive bacteria strains than against Gram-negative microorganisms (except for *P. aeruginosa*, which was resistant). Similar observations were demonstrated by Pereira et al. [47], who found no antibacterial efficacy of LGEO against *P. aeruginosa* and *Escherichia coli*. These findings contradict our results, showing a weak antibacterial activity of LGEO against *P. aeruginosa* (4.33 ± 0.58 mm). Generally, *P. aeruginosa* is well known for its high internal resistance to antiseptic and antibiotic agents, due to a low permeability of its outer membrane. Since the outer membrane limits the entry of the substances into the cell, Gram-negative bacteria are inherently resistant to hydrophobic antibiotics [55]. Based on all of the aspects, it can be postulated that the antibacterial effect of our LGEO tested in the current study can be related to the disturbances in the lipid fraction of bacterial plasma membranes, thereby causing changes in the membrane permeability with the consequent leakage of intracellular materials. Furthermore, the susceptibility and resistance of microorganisms is often determined by MIC, which is a parameter characterized as the minimum concentration of a substance able to suppress the bacterial growth [56]. Our findings re-

vealed that even highly resistant antimicrobial isolates, such as *P. aeruginosa* [47], showed sensitivity to LGEO, prognosticating its prospective use as a promising agent with the power to suppress the growth of a broad scale of bacterial strains. However, it is necessary to keep in mind that the effectiveness of EOs against the growth of bacteria and yeasts can be affected by diverse factors including bacterial status (resistance and susceptibility, persistence and tolerance), the size of inoculum and the antimicrobial concentrations used, too [57].

Commonly, microscopic filamentous fungi have a great ability to colonize various types of substrates, and grow even under extreme conditions. *Penicillium* spp. belong to the most considerable species producing the spoilage of foods [58]; therefore, these species were also selected in our research for analyses. In accordance with our study, Tzortzakis et al. [59] demonstrated that *Cymbopogon citratus* EO was able to reduce or completely inhibit the growth of tested fungal strains (*Colletotrichum coccodes*, *Botrytis cinerea*, *Cladosporium herbarum*, *Rhizopus stolonifer* and *Aspergillus niger)*, depending on the concentrations used. In addition, several studies have found that citral (the most abundant component in our EO) has strong antifungal effects against various *Penicillium* species including *P. digitatum* and *P. italicum* [60,61]. Fan et al. [62] also reported a dose-dependent suppression of the growth of *P. digitatum* induced by citral with a MIC of 2.0 µL/mL. Although the possible mechanisms essential for the antifungal action of citral have not been yet clarified, some probable machineries have been suggested. One of them is the disruption of cell integrity associated with cell organelle leakage and, ultimately, cell death, because of the lipophilic nature of the citral permitting the permeabilization of the cell membrane [63]. Furthermore, Rajput and Karuppayil [64] have shown that citral can elicit its antifungal activity via the inhibition of ergosterol biosynthesis, which is a principal constituent of the fungal cell membrane maintaining its structural integrity, fluidity and permeability [65].

The antibiotic resistance of bacteria can be provided by their ability to form cellular agglomerates, such as biofilms [66], as an avoidance reaction also contributing to persistence and virulence [67]. According to Coenye [68], biofilms are surface-attached and structured microbial communities that contain sessile cells immersed in a self-produced extracellular matrix, composed mainly of polysaccharides, but also DNA and other components. The formation of biofilms has been identified in diverse microorganisms, such as *Listeria monocytogenes* [69–71], *S. aureus* [72], *Pseudomonas* spp. [73,74] including *P. fluorescens* [75], as well as *S. enteritidis* [76]; the last two were also employed in our study. In relation to this, we have observed that LGEO was able to change the protein profile of the biofilms produced by *P. fluorescens* and *S. enteritidis*, indicating its anti-biofilm activity. Similarly, the results obtained by Adukwu et al. [72] revealed the anti-biofilm activity of *Cymbopogon flexuosus* EO against the antibiotic-resistant *S. aureus*. This activity was even greater than those of other EOs (obtained from grapefruit, lime, bergamot and lemon) investigated. In general, the inhibitory action of EOs on biofilm dispersal can be linked to hydrophobicity, reactivity and the EO diffusion rate into the matrix, as well as the structure and composition of the biofilms [77]. Taking into account the hydrophobic reaction as a fundamental contributor to the antimicrobial action against the bacteria [78], we hypothesize that the hydroxyl groups of active substances in our LGEO can directly interact with the extracellular polymeric substances of the biofilms, leading to hydrophobicity-induced changes in the mass spectra of the biofilms, formed by the two bacterial strains investigated. Furthermore, since the chemical profile of biofilms is considerably influenced by the metabolism of the biofilm microbiota [79], such an interaction can also participate in the protein profile alterations of both biofilms analyzed in our study. We assume that the high content of citral in our EO supported its anti-biofilm action. Regarding the results obtained from our in vitro studies, we further employed LGEO in our in situ analyses using selected food models to investigate the potential of the EO to be used in innovative food packaging.

Considering the promising potential of LGEO, we further investigated its antibacterial and antifungal efficacies using the vapor method on food models. Currently, researchers

focus on evaluating the antimicrobial potential of various EOs in the vapor phase, known for its stronger antimicrobial activity, in comparison with the liquid phase that uses direct contact [80]. In addition, the vapor phase permits the free attachment of EOs to microorganisms, while the lipophilic molecules present in the liquid phase usually disrupt a mycelium formation, thus resulting in the impossibility of EOs to attach to microorganisms [15].

Generally, it is known that EOs possess a valuable potential for use in the food industry, since they have an ability to effectively prevent the presence of some bacterial strains [81] including *S. marcescens* and *M. luteus*. These two bacterial strains, as common food spoilage bacteria [82,83], were evaluated in our in situ experiments, which showed the strong antibacterial efficacy of LGEO (in all concentrations used) against the growth of the bacteria on the bread model. A similar issue was also examined in the study by Vazirian et al. [84], who tested the growth suppression of selected bacteria (*B. cereus*, *C. albicans*, *E. coli*, *Salmonella typhimurium* and *S. aureus*) inoculated on cream-filled cakes after treatment with LGEO (1.0 L/mL). Their findings revealed that no microbial strain had grown on the cake after 72 h of storage, except for *S. aureus* that was resistant to LGEO. Thus, we can assume that although different methodologies were used in the studies, the results pointed out the promising antibacterial properties of LGEO applied to bakery products. This assumption can also be supported by Oliveira et al. [2], who analyzed the shelf-life of bread in experimental packages composed of cashew gum, gelatin, ferulic acid and lemongrass EO. They found that the loaves of bread wrapped with the addition of LGEO had a high proportion of citral in their crust, and they exhibited a longer shelf-life (6 days) as compared to conventional commercial bread (3 days). Hence, this finding indicates the potency of the LGEO antimicrobial effect, which is also supported by our findings.

The composition of the food system (availability of nutrients for microorganisms) impacts the antimicrobial efficacy of EOs, and this activity is usually decreased in in situ experiments in comparison with in vitro ones. However, the low fat content of vegetables can participate in the successful results obtained with the application of EOs, as reported by Burt [44]. Hence, in our research paper, this effectiveness has been investigated for carrots and celery (as food model substrates). The antibacterial activity of LGEO against *Salmonella enterica* inoculated on different types of leafy greens (iceberg and romaine lettuces, and spinach) was determined by Moore-Neibel et al. [85]. From this experimental work it is evident that the antibacterial efficacy of lemongrass EO was dose-dependent, which is consistent with our findings. Moreover, we propose that the different inhibitory actions of LGEO on the growth of *M. luteus* and *S. marcescens* inoculated on carrot and celery, in our experiment, can be associated with the variations in the chemical composition of vegetable substrates and the availability of nutrients for the growth of microorganisms.

Furthermore, an inhibitory action of LGEO on the mycelia growth of the analyzed *Penicillium* spp. was (as in the case of in vitro analyses) also observed in the vapor method. Indeed, despite the high resistance of *Penicillium* strains [25], their growth was suppressed by the EO on all the food models with the strongest effectiveness at the highest concentration (500 μL/L). The findings are in agreement with the research of Mani-López et al. [86], who evaluated the antifungal impact of LGEO in the vapor phase on the growth inhibition of *P. expansum* inoculated on the bread samples. The authors determined the inhibitory effect of LGEO in a concentration of ≥750 μL/L (after 21 days of sample incubation), which increased with the increasing concentration. The antifungal activity of LGEO used in the current study supported our previous studies dealing with the in situ antifungal efficacy of other EOs, such as *Thymus serpyllum* [23] or *Syzygiumaromaticum* [24], against the *Penicillium* spp. growing on bread and carrots. From the results of all the analyses it can be noted that LGEO appears to be a promising component with a potential application to extend the shelf-life of bakery products and vegetables in the food industry. However, it is commonly known that LGEO (mainly citral) has a characteristic aroma; thus, the determination of the impact of LGEO concentrations on the sensory properties of the food products is the next challenge of our experiments.

## 5. Conclusions

In the current study, the antimicrobial (in vitro, in situ), antibiofilm and antioxidant activities, as well as the chemical composition of commercial LGEO obtained from the Slovak company (Hanus Ltd., Nitra, Slovakia) were investigated. Our findings revealed the strong antioxidant activity of the EO tested with citral, geraniol and 1,8-cineole being the principal compounds of its chemical composition. The in vitro antimicrobial evaluation showed that LGEO was effective in inhibiting the growth of a wide range of tested microorganisms, and its effectiveness mainly depended on its concentration. We noticed its inhibitory action even against resistant microorganisms. Furthermore, it was observed that *Cymbopogon citratus* EO influenced the mass spectra of the biofilms produced on various surfaces, as it was detected by the MALDI-TOF MS Biotyper. Its antimicrobial potential was also recorded in in situ experiments. Additionally, the obtained results suggest that the incorporation of LGEO as a natural antimicrobial agent into the active packaging of food products (including bakery products and vegetables) has a perspective to prolong their shelf-life. The great benefit for the food industry mainly lies in the easy extraction of the EO, its environmental friendliness and its large portfolio of proven biological functions, which is also documented in our study. To clarify the usage of LGEO as an antimicrobial agent, we plan to include sensory analyses that would reveal what concentration of LGEO is acceptable to product consumers. In addition, these data complement our previous research, providing an extensive overview of the biological functions of several commercial EOs purchased from the Hanus company.

**Author Contributions:** Conceptualization, V.V., H.Ď. and M.K.; methodology, V.V., H.Ď., L.G., P.B., N.L.V., M.V. and M.K.; software, V.V. and M.K.; validation, V.V., L.G. and M.K.; formal analysis, V.V., H.Ď. and M.K.; investigation, V.V., H.Ď., L.G., P.B., N.L.V., M.V. and M.K.; resources, M.K.; data curation, V.V. and M.K.; writing—original draft preparation, V.V., H.Ď. and M.K.; writing—review and editing, V.V., H.Ď. and M.K.; visualization, V.V.; supervision, M.K.; project administration, M.K.; funding acquisition, M.K. All authors have read and agreed to the published version of the manuscript.

**Funding:** This research was funded by the grant APVV-20-0058 "The potential of the essential oils from aromatic plants for medical use and food preservation".

**Data Availability Statement:** Not applicable.

**Acknowledgments:** This work was supported by the grants of the VEGA no. 1/0180/20.

**Conflicts of Interest:** The authors declare no conflict of interest.

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
