# Peer review of "Cymbopogon citratus Essential Oil: Its Application as an Antimicrobial Agent in Food Preservation"

_agronomy, doi:10.3390/agronomy12010155_

Round 1

Reviewer 1 Report

The paper investigates the antimicrobial and antioxidant activity of LGEO. The paper is well written and is worthy of publication. However, in order to improve the manuscript, the following comments and suggestions should be considered and addressed.

The authors rightly attributed the highest antimicrobial activity to two isomers: geranial (citral A) and neral (citral B). Line 360. On the other hand, the antimicrobial activity of isolated citral has been investigated in previous work, e.g.

Antimicrobial activity of geraniol: an integrative review https://doi.org/10.1080/10412905.2020.1745697

Inhibitory efficacy of geraniol on biofilm formation and development of adaptive resistance in Staphylococcus epidermidis RP62A

DOI:10.1099/jmm.0.000570

It would be useful to compare the antimicrobial activity of geranial versus LGEO. This would enable an analysis of the effect of the other components present in lemongrass essential oil.

Can other oil ingredients, present in small quantities, have a synergistic effect?

Author Response

Reviewer #1

The paper investigates the antimicrobial and antioxidant activity of LGEO. The paper is well written and is worthy of publication. However, in order to improve the manuscript, the following comments and suggestions should be considered and addressed.

Response: Thank you very much for such a favorable opinion.

Point 1: The authors rightly attributed the highest antimicrobial activity to two isomers: geranial (citral A) and neral (citral B). Line 360. On the other hand, the antimicrobial activity of isolated citral has been investigated in previous work, e.g.

Antimicrobial activity of geraniol: an integrative review https://doi.org/10.1080/10412905.2020.1745697

Inhibitory efficacy of geraniol on biofilm formation and development of adaptive resistance in Staphylococcus epidermidis RP62A DOI:10.1099/jmm.0.000570

It would be useful to compare the antimicrobial activity of geranial versus LGEO. This would enable an analysis of the effect of the other components present in lemongrass essential oil.

Can other oil ingredients, present in small quantities, have a synergistic effect?

Response: Thanks for the inspiring question. Of course, we assume that the presence of other components (e.g. geraniol and 1,8-cineole as reported in our manuscript) of LGEO may also affect its antimicrobial activity. Moreover, as we state in the manuscript, also the antioxidant capacity of LGEO depends mainly on its chemical composition, and the activity is not attributed to just one or only some of its components but it is a result of the synergistic effect of all its constituents with different bioactivities, functional groups and polarities.

We will certainly consider the analysis of geranial itself against LGEO in our further research, in which we plan to continue in the future to expand the findings in this area.

Reviewer 2 Report

The subject of the manuscript is topical. While there are investigations on the composition and activity of essential oil from the same plant, the length, quality and language of the paper are adequate. The used methods are accurate. The bacterial strains used in the study are also different form the ones used by other authors, which contributes the current knowledge about this essential oil and its potential practical applications.

I have only one recommendation to authors: please add additional information about identification of components of essential oils, chemical family, retention index (RI) and retention time (RT) in Table 1.

Author Response

Reviewer #2

The subject of the manuscript is topical. While there are investigations on the composition and activity of essential oil from the same plant, the length, quality and language of the paper are adequate. The used methods are accurate. The bacterial strains used in the study are also different form the ones used by other authors, which contributes the current knowledge about this essential oil and its potential practical applications.

Point 1: I have only one recommendation to authors: please add additional information about identification of components of essential oils, chemical family, retention index (RI) and retention time (RT) in Table 1.

 Response: Thank you very much for such a favorable review. It has been corrected.

Reviewer 3 Report

The manuscript deals with the Cymbopogon citratus essential oil: its application as antimicrobial agent in food preservation. The manuscript is interesting. The following article should be referred: “Ekpenyong, C. E., & Akpan, E. E. (2017). Use of Cymbopogon citratus essential oil in food preservation: Recent advances and future perspectives. In Critical Reviews in Food Science and Nutrition (Vol. 57, Issue 12, pp. 2541–2559). Informa UK Limited. https://doi.org/10.1080/10408398.2015.1016140”

The English language must be revised.

Abstract

Line 28- “Our results suggest LGEO to be a promising natural antimicrobial agent applied in active food packaging.”??in which packaging material??and suitable for which type of food?

Introduction

The topics must be better linked.

Materials and methods

Please add each equation separated from the text and numbered.

Line 104- “The AA values increased in the following manner: weak (0 – 29%) < medium-strong (30 – 59%) < strong (60 and more %). Moreover, the value for total AA was expressed according to the calibration curve as 1 µg of standard reference Trolox to 1 mL of the GMEO sample (TEAC).”??calibration curve? equation??coefficient of determination??

Results and discussion

Pictures of each sample???

This section must be improved. The authors must explain better the reason(s) of the obtained results.

Conclusion

Please do not repeat your results and focus on your main conclusions.

Line 522- “Anyway, the obtained results suggest that incorporation of LGEO as a natural antimicrobial agent into active packaging of food products has a potential for extending their shelf-life. In addition, these data complement our previous research to provide a comprehensive overview of the biological functions of various commercial Eos obtained from the Hanus company.”??any disadvantages??undesired flavors??

References

Around 46 references have more than 5 years. Please update your list of references.

Line 566- Please format the scientific names in italic “Thymus vulgaris”.

Author Response

Reviewer #3

Point 1: The manuscript deals with the Cymbopogon citratus essential oil: its application as antimicrobial agent in food preservation. The manuscript is interesting. The following article should be referred: “Ekpenyong, C. E., & Akpan, E. E. (2017). Use of Cymbopogon citratus essential oil in food preservation: Recent advances and future perspectives. In Critical Reviews in Food Science and Nutrition (Vol. 57, Issue 12, pp. 2541–2559). Informa UK Limited. https://doi.org/10.1080/10408398.2015.1016140”

Response: Thank you for the favorable opinion. The reference is added directly in the manuscript.

Point 2: The English language must be revised.

Response: The English language has been modified directly in the manuscript.

Point 3: Abstract

Line 28- “Our results suggest LGEO to be a promising natural antimicrobial agent applied in active food packaging.”??in which packaging material??and suitable for which type of food?

Response: Edited directly in the manuscript.

Point 4: Introduction

The topics must be better linked.

Response: Edited directly in the manuscript.

Point 5: Materials and methods

Please add each equation separated from the text and numbered.

Line 104- “The AA values increased in the following manner: weak (0 – 29%) < medium-strong (30 – 59%) < strong (60 and more %). Moreover, the value for total AA was expressed according to the calibration curve as 1 µg of standard reference Trolox to 1 mL of the GMEO sample (TEAC).”??calibration curve? equation??coefficient of determination??

Response: Edited directly in the manuscript.

Point 6: Results and discussion

Pictures of each sample???

This section must be improved. The authors must explain better the reason(s) of the obtained results.

Response: Pictures of samples have been added. Revised directly in the manuscript.

Point 7: Conclusion

Please do not repeat your results and focus on your main conclusions.

Line 522- “Anyway, the obtained results suggest that incorporation of LGEO as a natural antimicrobial agent into active packaging of food products has a potential for extending their shelf-life. In addition, these data complement our previous research to provide a comprehensive overview of the biological functions of various commercial Eos obtained from the Hanus company.”??any disadvantages??undesired flavors??

Response: Revised directly in the manuscript.

Point 8: References

Around 46 references have more than 5 years. Please update your list of references.

Line 566- Please format the scientific names in italic “Thymus vulgaris”.

 Response: Revised directly in the manuscript.

Round 2

Reviewer 3 Report

Please read the manuscript carefully and perform all recommended changes.

The English language must still be revised, e.g. Line 670- “Anyway, the obtained results suggest that incorporation of LGEO as a natural antimicrobial agent into active packaging of food products (including bakery products and vegetables) has a perspective to prong their shelf-life.”??prong?

References

Please format all scientific names in italic, e.g. lines 737 and 746.

Author Response

Reviewer #3
Point 1:  Please read the manuscript carefully and perform all recommended changes.
The English language must still be revised, e.g. Line 670- “Anyway, the obtained results suggest that incorporation of LGEO as a natural antimicrobial agent into active packaging of food products (including bakery products and vegetables) has a perspective to prong their shelf-life.”??prong?
Response: It was corrected.

Point 2: Please format all scientific names in italic, e.g. lines 737 and 746.

Response: It was corrected.